# Do Intraoperative Platelet-Rich Plasma Injections Influence the Final Appearance of Vertical Scars after Breast Reduction? Spectrophotometric Analysis

**DOI:** 10.3390/jcm13030691

**Published:** 2024-01-25

**Authors:** Tomasz Zawadzki, Aneta Sitek, Bogusław Antoszewski, Anna Kasielska-Trojan

**Affiliations:** 1Plastic, Reconstructive and Aesthetic Surgery Clinic, Institute of Surgery, Medical University of Lodz, 90-153 Lodz, Poland; tzawadzki2233@gmail.com (T.Z.); boguslaw.antoszewski@umed.lodz.pl (B.A.); 2Department of Anthropology, University of Lodz, 90-237 Lodz, Poland; aneta.sitek@uni.lodz.pl

**Keywords:** scar, platelet-rich plasma, dermospectrophotometry

## Abstract

**Background**: Platelet-rich plasma (PRP) has been shown to support wound healing and tissue regeneration due to its high concentration of growth factors and cytokines. This study aims to investigate the effect of intraoperative PRP injections on the final appearance of vertical scars after breast reduction, as well as to identify potential predictors of a scar’s aesthetic assessment using spectrophotometric parameters. **Methods**: In this prospective, randomized trial, 82 scars from 41 women who underwent bilateral breast reduction with an inverted T pattern were analyzed. PRP or a placebo substance (0.9% sodium chloride solution) was injected intraoperatively into the edges of vertical wounds. Spectrophotometric measurements of scar pigmentation were performed 3 and 6 months after surgery; additionally, two independent observers evaluated the aesthetic appearance of scars based on photographs. **Results**: The results showed that the use of intraoperative PRP injections did not significantly influence the final appearance of vertical scars after breast reduction. **Conclusions**: We indicated spectrophotometric variables (b) in the early stages of wound healing (after 3 months) that can be predictors of the final scar’s aesthetic outcome. This can be helpful in detecting scars that may need additional interventions to optimize the healing process.

## 1. Introduction

Breast reductions are in 10th place among worldwide plastic surgery procedures (according to ISAPS, 2021 [1]), with statistics displaying a constant upward trend. The multitude of surgical techniques, as well as several general and local factors that may influence the result, mean that breast surgeons are in constant pursuit of improving postoperative results in different aspects: breast shape and symmetry, nipple–areolar complex (NAC) sensation, and the appearance of scars. Scars are a very important aspect of the final aesthetic result. There are many methods of breast reduction regarding the shape and location of scars on the chest. The most commonly used resection scheme is the “inverted T” pattern proposed by Wise in 1956 [2]. In later years, Lassus and then Lejour popularized methods to reduce the presence of scars in the inframammary folds [3,4]. Further modifications include the use of different pedicles for the transposition of the nipple–areolar complexes, while the arrangement of the postoperative scars remains unchanged [5,6]. In many breast surgeries (both aesthetic and reconstructive), vertical scars remain permanent and are the most visible regardless of the extent of surgery (breast reduction, mastopexy, augmentation mastopexy). As a response to the problem of vertical scars, Lalonde (2003) proposed a no-vertical-scar breast reduction technique [7]. However, this technique is not widely used, mainly because of the long learning curve and the postoperative tendency toward flat breasts.

Bearing in mind that vertical scarring is “unavoidable” in plastic/breast surgeons’ practice, many authors have focused on methods of appearance enhancement. Surgeons, as well as patients, look for various available treatments in the field of plastic surgery, aesthetic medicine, and physiotherapy, including combined therapies, to obtain the most aesthetically desirable postoperative scars [8,9,10]. These techniques influence the different stages of wound healing. Healing involves a multifaceted process governed by sequential yet overlapping phases, including the hemostasis/inflammation phase, proliferation phase, and remodeling phase. After an injury to the skin, the inflammatory phase begins when the exposed sub-endothelium, collagen, and tissue factors activate platelet aggregation, which results in degranulation and the release of chemotactic factors (chemokines) and growth factors (GFs) to form a clot and achieve successful hemostasis. A further course of inflammation is activated by platelet-derived cytokines. The fact that platelets initiate the healing process encouraged attempts to use their potential in aesthetic and regenerative medicine. One of the most popular treatments since the 1990s has been therapy with platelet-rich plasma (PRP). This is an autologous concentrate of platelets of 4–7 times higher concentration than normal blood and plasma. It has been proven to be effective in the treatment of several medical conditions, including post-traumatic wounds, ulcers, chronic wounds, orthopedic injuries to the cartilage and/or tendons, and many others. Its availability and easy processing, full histocompatibility, and low risk of complications make it an attractive therapeutic option for healing improvement [11,12]. The exact mechanism of action of PRP is not yet fully understood, but it is known that PRP contains increased quantities of cytokines, chemokines, growth factors, and a fibrin scaffold, which provides more rapid hemostasis and may lead to more rapid tissue regeneration [11,12,13,14,15].

The aim of this study is to examine the effect of intraoperative injections of platelet-rich plasma (PRP) on the final appearance of vertical scars after breast reduction and to verify spectrophotometric parameters as eventual predictors of a scar’s aesthetic assessment.

## 2. Materials and Methods

In this prospective, randomized trial, we analyzed 82 scars of 41 women aged 18–60 who underwent bilateral breast reduction with an inverted T pattern (between 2019 and 2022) at a plastic, reconstructive, and aesthetic surgery clinic. 

We only included those patients for whom the surgical procedure was performed by one surgical team with the same surgical technique—reduction mammaplasty with nipple–areolar complex transfer on a dermal superomedial pedicle with inverted T skin resection (Wise pattern), who had no comorbidities which could impact wound healing (three women had well-controlled hypertension), and who gave written informed consent to participate in the study.

We excluded all cases where any complications appeared (three women—wound dehiscence (one in the middle of a vertical scar and two in both T junctions, over 5 mm) and two women—nipple–areolar complex necrosis (left-sided, including about 1/3 of the upper pole of the areola, treated with debridement and iodine dressing (Inadine)) and patients who did not follow postoperative recommendations and/or did not attend follow-up visits (four women). Delayed wound healing in the T junction without wound dehiscence over 5 mm was not considered a complication due to its regular occurrence. In such cases, iodine dressing (Inadine) was administered for 7 days (with the recommendation of a daily change), and then, triple antibiotic ointment was recommended for the final epithelialization of the wound. In all cases, during a follow-up visit after 2 weeks of such treatment, T junction wounds were closed and received routine recommendations. All participants were given the same postoperative recommendations and declared full compliance with them, including using greasing ointment twice a day (a day after suture removal) and silicone gel (from the same manufacturer) two weeks after suture removal for at least 6 months. Direct anthropometric measurements were taken from each study participant before surgery: height, weight, distance from the sternal notch to the nipple (SN-N), and the length of the planned vertical incision. At the end of the surgery (after the completion of suturing—subcutaneous sutures and intradermal sutures with Monosyn 3.0 (Braun)—before the application of sterile stripes), each patient’s blood sample (8cc) was collected by an anesthesiologic nurse from new intravenous access to a commercial kit used to obtain PRP with ACD-A anticoagulant (KeyMed, Jelenia Góra, Poland). The PRP tube was centrifugated according to the kit’s manufacturer guidelines prepared for our device (Zenithlab LXG-A Centrifuge, 1120 rounds per minute/5 min). The obtained platelet-rich plasma was used for alternating intradermal injections every 5mm into the edges of the vertical wound from the lower edge of the nipple–areola complex to the inframammary fold (Figure 1). The vertical scar on the other breast was injected with a 0.9% solution of NaCl in the same manner and in the amounts corresponding to the injected platelet-rich plasma. Injections were performed with a 1 mL syringe and with the use of a 25G needle. The PRP-injected side was selected based on the outcome of a coin toss by the nurse, which was performed after the completion of suturing, with heads representing the right side and tails the left side. Information about the side subjected to injections and the amount of platelet-rich plasma was concealed from the patient during every stage of the clinical trial, but it was recorded in the study protocol. PRP injection was only performed once, intraoperatively. During follow-up visits in the outpatient clinic, 3 and 6 months after the surgery, the pigmentation of the middle point of the vertical scar was measured using a DSM II (Cortex Technology, Aalborg, Denmark) dermospectrophotometer (the average of three separate measurements was used for statistics), and a set of standardized photographic documentation was made (photos in three projections—en face, left, and right profile) (Figure 2). The periods of 3 and 6 months after surgery for spectrophotometric examinations were chosen due to the following aspects: (1) most of the absorbable sutures used dissolve within 3 months, so after this time, they should not have affected the result of the examination and (2) 6 months after the surgery, the patients were free to use other techniques to diminish the scars’ appearances.

### 2.1. PRP Quantification

Compared with the mean baseline platelet count in the whole blood collected with ethylenediaminetetraacetic acid (EDTA), the mean quantification of platelets in the PRP collected with the tube with anticoagulant ACD-A (citrate dextrose solution A) and centrifuged according to the manufacturer’s guidelines increased by about 2.2 times for the mean volume of PRP obtained in our study (2.7 mL) (based on the data provided by the manufacturer Keymed, Poland, on our request). Given the mean platelet count in our sample (in all patients who qualified for the surgery the platelet count was within normal range, collected for EDTA before surgery)—0.33 × 10^6^/µL—the mean concentration of platelets in our PRP was estimated to be 0.65 × 10^6^ platelets/µL, with the minimum value of 0.59 × 10^6^ platelets/µL and maximum value of 0.84 × 10^6^ platelets/µL. Based on the related references, this count can be accepted as representative for PRP [16,17].

After the completion of the study, an independent team of observers (unfamiliar with the aim of the study), a plastic surgeon (observer 1), and a gynecologist (observer 2) were asked to evaluate the aesthetic appearance of the scars presented in photographs. Their subjective assessment of an isolated fragment of photographs presenting vertical postoperative scars (to eliminate the suggestion of the appearance of the entire breast) was based on photographic documentation made 6 months after the surgery using a 5-point Likert scale, where 1—unaesthetic, very visible (red, wide, hypertrophic) and 5—almost invisible (skin-color, thin) (Figure 3). 

The study protocol was approved by the Bioethics Committee of the Medical University of Łódź (RNN/366/19/KE).

### 2.2. Statistical Analysis

The normality of distributions of the variables was tested with the Shapiro–Wilk test. In the case of variables for which the distribution deviated from normality and further analyses were required to meet such an assumption, the Box–Cox (B-C) transformation was applied. The weight of the breast tissue removed from the breast injected with PRP and NaCl was compared with Student’s t-test. Agreement on the subjective scores between observers was assessed with the agreement coefficient (alpha Krippendorf). The Mann–Whitney statistic corrected for ties was used to compare scars’ appearance scores between the sides (PRP vs. NaCl). The spectrophotometric parameters of scars 3 and 6 months after the procedure, depending on the side (PRP vs. NaCl), were compared using a two-factor ANOVA with the interaction effect. The influence of the density of the injections of PRP/NaCl on the assessment of scars was examined using Spearman’s rank correlation. The analysis of relationships between the mean assessment of the scar (dependent variable) and the age and BMI of the patients, the weight of the removed breast tissues, the length of the vertical scar, the density of the PRP/NaCl injections along the wound (mL/cm), and the spectrophotometric parameters of the scar 3 and 6 months after the procedure in interaction with the type of injection (PRP or NaCl) was performed using the general linear model (GLM)—two-factor ANOVA with the interaction effect. Multivariate regression was used to determine variables that influenced the scar assessment. In the first step, a full model (including all analyzed independent variables) was generated, and then, a stepwise model was built (including only significant and non-redundant variables). Logistic regression was used to establish a model predicting scars with a mean score after 6 months of at least 4.5. The final model was designed using the backward step method. The quality of the classifier was illustrated with the ROC curve. All analyses were performed in STATISTICA version 13.3.

## 3. Results

Table 1 summarizes the basic characteristics of the variables included in the study. Table 2 presents the distribution of the scores obtained by scars injected with PRP and a placebo 6 months after the procedure. The agreement coefficient (alpha Krippendorf) for the scars injected with PRP was 0.91, and for the scars injected with the placebo, it was 0.82. The Krippendorf alpha coefficient for all scores (both PRP and the placebo) was 0.86. All the above results exceeded the value of 0.80 and therefore indicated very good agreement between the judges’ assessments. This made it possible to average these scores and use the average values in further analyses.

The weight of the gland removed on the side injected with PRP and on the side injected with NaCl did not differ significantly (*p* = 0.88). The assessment of scars on both sides was also similar, indicating no effect of PRP on the final appearance of the scar (*p* = 0.996) (Table 2).

### 3.1. Spectrophotometric Parameters 3 Months versus 6 Months after Surgery, PRP vs. NaCl

Most of the spectrophotometric parameters of the scar changed between 3 and 6 months after the procedure, but the size and direction of this change were not related to the type of injection (PRP vs. placebo). These changes involved all three scar color coordinates in the RGB system: the L parameter (CIELab), the melanin index, and the erythema index (Table 3). The values of the coordinates R, G, B, and L increased, which indicated the lightening of the scar. This was further reflected by the reduction in the melanin index. The erythema index also decreased. All these changes occurred similarly on the side injected with PRP and NaCl. Only parameters a and b (CIELab) did not change in the analyzed periods. This applied to both the PRP- and NaCl-injected scars (Table 3). 

Although the assessment of scars (scores) did not differ between scars injected with PRP and NaCl, Spearman’s rank correlation showed that the scores of scars injected with PRP correlated positively with the density of PRP injections (R = 0.32, *p* = 0.044), while on the NaCl-injected side, such a correlation was not found (R = 0.05; *p* = 0.78).

### 3.2. General Linear Model (GLM)—Variables Influencing Scars’ Scores

Using two-factor ANOVA with the interaction effect, it was determined whether the type of injection (PRP vs. NaCl) modified the correlation between the assessment of scars 6 months after the procedure and each of the independent variables. It was found that the assessment of the scar depended on the length of the scar, the weight of the removed tissues (negative correlation), and the spectrophotometric parameters—R, G, L, and b—measured 3 months after the procedure (positive correlation; a lower intensity of blue color means higher values of coordinate b). The relationships did not depend on the type of injection (Table 4).

### 3.3. Multivariable Analysis

In the next step, we performed a multivariable analysis to explain the variability in the assessment of scars (after 6 months). The model including all variables appeared not to be significant (*p* = 0.089), although three variables (body weight, height, and BMI) were significantly associated with the assessment of scars. As many independent variables in the model were redundant (because of strong interrelation), in the next stage, we built a stepwise model (limited to significant, not inter-related variables). It only included one variable (R^2^ = 0.1005, corrected R^2^ = 0.0893; F(1.80) = 8.94 *p* < 0.0037; error: 0.67) that correlated with the final assessment of a scar—vertical scar length (b = −0.32, error for b = 0.106, t = −2.99, *p* = 0.004). This variable explained about 10% of the assessment variability (Table 5).

### 3.4. Logistic Regression Model—Predictors of a Very Good Score in Scar Assessment (>4.5)

In the last stage of the analysis, the scars were divided into two categories: those with an average score of at least 4.5 on the Likert scale and others (rated lower). Then, using the backward stepwise method, a logistic regression model was designed to differentiate both categories of scars. All variables that significantly correlated with the assessment of scars in two-factor ANOVA (GLM) were introduced into the model. Due to collinearity, most of these variables lost significance in the regression analysis. Then, the least significant factors were removed until a model consisting only of significant predictors was obtained. Finally, one significant predictor of very good assessment (>4.5) remained—spectrophotometric parameter b (CIELab) measured 3 months after the procedure (*p* = 0.02, OR = 1.27, 95% CI 1.04–1.55). The ROC curve showed that coordinate b, as an independent factor, did not provide a satisfactory classification (AUC = 0.654), but it significantly contributed to the differentiation of scars within the selected categories (*p* = 0.024). The optimal cut-off point for this parameter, determined by the Youden coefficient, was 6.34 (a b value higher than this value (including fewer blue components) predicted a score > 4.5). The sensitivity and specificity for the model were 0.46 and 0.86, respectively (Figure 4)

## 4. Discussion

Scars are an inseparable consequence of any surgical intervention in the human body. Visible scars may be responsible for an unsatisfactory aesthetic result in surgery. In many breast-related procedures, e.g., reduction mammoplasties, scars are extensive and often located on the visible curvatures of the breast. For this reason, plastic surgeons have been looking for the best techniques that leave the least visible skin scars (regarding their localization and extent). Although there are many available techniques, none are both universal for every patient and allow satisfactory breast shapes with minimal scarring. White et al. (2013) conducted a prospective scar placement preference questionnaire with patients planned for gigantomastia surgery and found a significant predominance of patients’ preference for the “no vertical scar” technique popularized by Lalonde [7,18]. However, these preferences appeared to change when patients filled in questionnaires retrospectively after surgery. Sprole et al. (2007) compared different components of postoperative scars in patients undergoing breast reduction using the Wise technique. The results confirmed patients’ dissatisfaction with the location of the vertical scar, and a high rate of respondents who declared their willingness to remove it (50%) was reported. However, horizontal scars correlated with the most troublesome symptoms, e.g., pain or itching (65% of the respondents) [19]. Similar conclusions were presented by Celebirel et al. (2005), who reported that patients’ subjective assessment favored periareolar scars, while inframammary scars were assessed as the least pleasant, and vertical scars remained the most problematic for surgeons [20]. Our results also showed that the length of a vertical scar correlated with its aesthetic assessment in the multivariable analysis. In such a complex problem, both patients and surgeons are increasingly looking for additional methods to support healing that will result in “aesthetic”, almost invisible scars.

Refahee et al. (2020) designed a study to verify the influence of the perioperative injection of PRP in wound edges on scars’ aesthetics. The authors included 24 children treated surgically for complete unilateral cleft palates; 12 of them received intraoperative PRP injections, and the remaining 12 constituted the control group (no additional intervention). A statistically significant reduction in the width of the postoperative scar was observed in the study group [21]. These results should, however, be interpreted with caution, and the following limitations should be mentioned: individual variability and a lack of “true” controls (e.g., injection with NaCl to eliminate the effect of micro-needling). In our study design, the patient herself was a control, which allowed us to eliminate the effect of genetic differences, and the control wounds (on the other breasts) were subjected to analogous injections of 0.9% NaCl solution to exclude the effect of micro-needling. According to a meta-analysis conducted by Juhasz et al. (2020) on 58 studies (1845 patients), positive effects of micro-needling on healing were demonstrated [9]. In our study, we also analyzed the effect of intraoperative PRP injection, but, as well as the subjective evaluation method, we also used spectrophotometric parameters to evaluate the color and pigmentation of the examined and control scars. Moreover, we analyzed these parameters in two follow-up periods (3 and 6 months after the procedure) and controlled for changes within this period. 

Dermospectrophotometry has already been used in some studies as a tool for the assessment of scars [22,23]. Van der Wal et al. (2013) showed a correlation of scars’ appearances with the erythema index (with the E parameter corresponding with vascularity), measured using the DSM II tool. The authors assessed 50 scars using three tools: the Mexameter, Colorimeter, and DSM II tools, and showed a good correlation of their reads with the mean Patient and Observer Scar Assessment Scale (POSAS) scores assessed by two clinicians for all of the analyzed devices [24]. However, the literature concerning dermospectrophotometric analysis in scar evaluation is scarce, and these variables have been used much more often as predictors of the occurrence of skin cancer. Sitek A. et al. (2016) found the R coordinate of the RGB and the melanin index for skin on the buttock as predictors of skin cancer [25]. Fijałkowska M. et al. (2022) confirmed that the melanin index has predictive value when measured on the upper arm or buttock, as well as the erythema index’s relation to some antimicrobial peptide levels (cathelicidin and beta-defensin-2) [26,27]. To date, no studies have analyzed spectrophotometric parameters as predictors of the aesthetic appearance of scars or used these parameters as a measure of the efficacy of the intervention. Logistic regression showed that one of the spectrophotometric parameters—b, measured 3 months after surgery—appeared to be a predictor of scars’ appearances after 6 months. A parameter b value lower than the cut-off value (with a higher intensity of blue color) predicted a mean evaluation score that was less than “very good”. This can be referred to as the worse vascularity (more cyanosis) of scars after 3 months which predicts worse healing and a worse final appearance for the scar. This may be used as an easy tool to detect wounds and scars in the early stage of healing (3 months after the procedure) that should be subjected to additional interventions to enhance wound vascularization and healing (micro-needling, lasers, mesotherapy, etc.) to ensure the optimal final appearance of the scar.

The study has some limitations. We only included Caucasian women; thus, the results may be ethnically specific and may not allow for generalization. Our study group only included non-smoking women, so the results were not controlled for this characteristic. Also, the observer evaluations of scars using the 1–5 Likert scale can be regarded as an additional limitation, as this is not objective and was not validated for this purpose. However, our aim was to look for differences in the assessment of scars, in addition to objective changes described by spectrophotometric parameters, not to evaluate scars in detail. Furthermore, our preliminary observation (the model predicting the appearance of scars in the long-term observation) should be verified on a larger number of participants, and the model should be validated on an external sample. It would be worth extending the protocol of the presented study and other periods for spectrophotometric examinations, especially to include a longer follow-up period, e.g., 12 months. Additionally, the protocol of the study only included a single intraoperative injection of PRP, but it would be worth extending the protocol to include multiple injections.

## 5. Conclusions

In summary, it can be concluded that the main predictors of the appearance of vertical scars after breast reductions are related to the extent of the surgery, while the use of intraoperative PRP injections does not influence the scars’ final appearance. However, the density of PRP injections (mL/cm) seems to correlate with the postoperative score (contrary to the control scars injected with the placebo substance), which indicates the need for further investigations, as this observation is not related to micro-needling itself. Dermospectrophotometry can be a useful tool to assess postoperative scars’ evolution through the healing process. Spectrophotometric variables (b) in the early stage of wound healing may be a predictor for the less aesthetic final appearance of scars and can indicate the need for additional intervention to optimize the healing process. More research is needed in a larger and more clinically diverse study group to verify these observations.

## Figures and Tables

**Figure 1 jcm-13-00691-f001:**
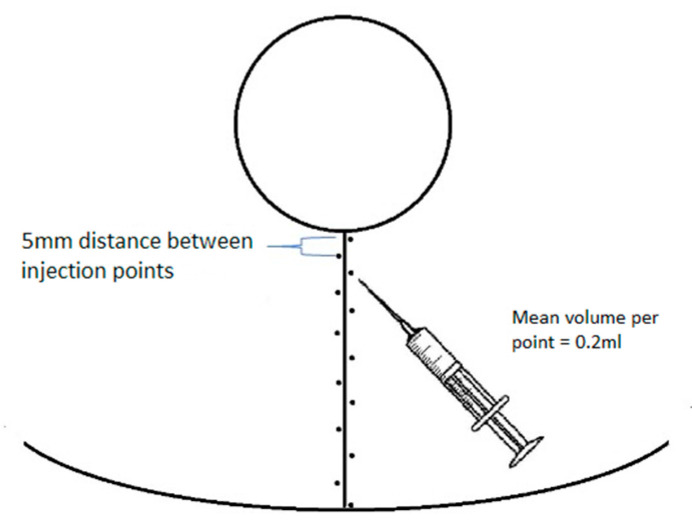
Scheme of PRP/NaCl injection points in vertical wound edges.

**Figure 2 jcm-13-00691-f002:**
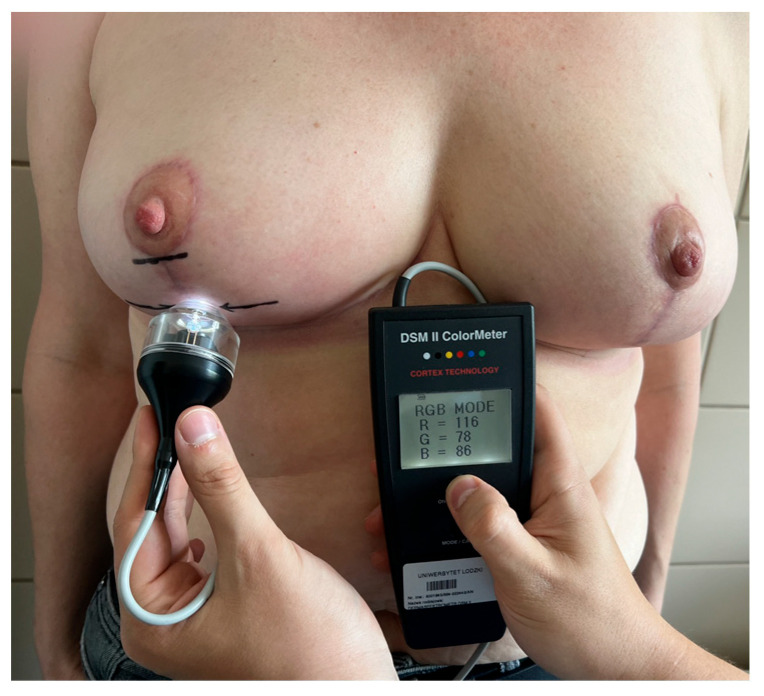
Pigmentation of the middle point of the vertical scar component measured using a DSM II (Cortex Technology, Aalborg, Denmark) dermospectrophotometer. (Arrows represent the middle point of the vertical scar).

**Figure 3 jcm-13-00691-f003:**
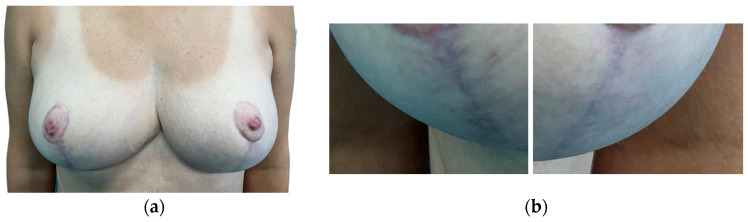
Subjective assessment of isolated parts of photographs presenting vertical postoperative scars. (**a**) Full standardized photography 6 months after surgery. (**b**) Isolated part of scars for observers’ assessment.

**Figure 4 jcm-13-00691-f004:**
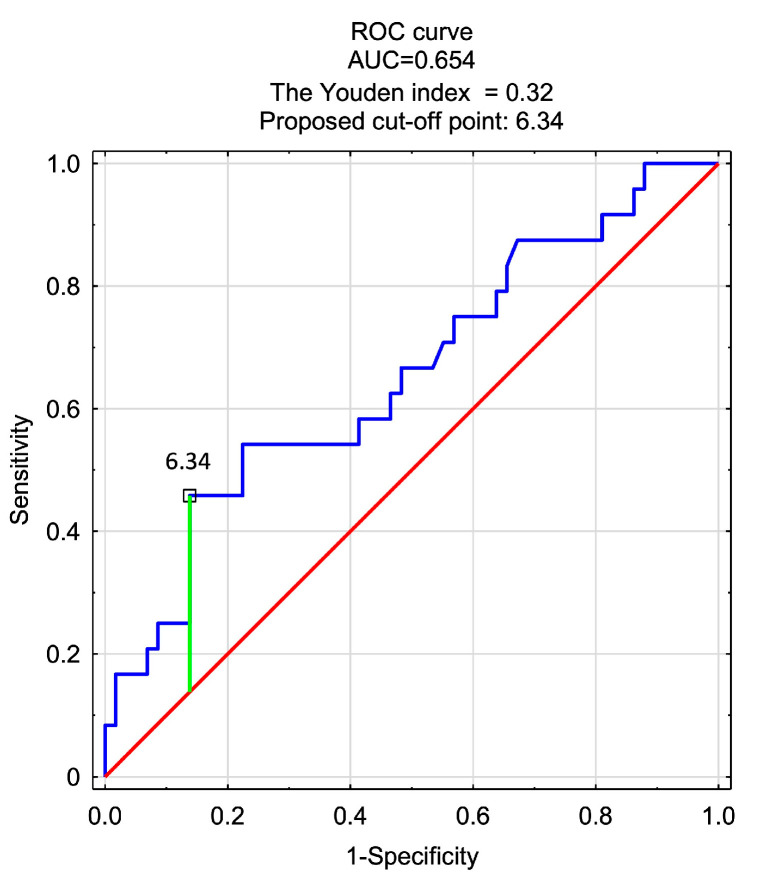
ROC curve showing the classification quality of spectrophotometric parameter b (CIELab) measured three months after surgery for the assessment of scars six months after surgery ((1) score ≥ 4.5; (0) rating < 4.5).

**Table 1 jcm-13-00691-t001:** Characteristics of the analyzed variables.

Variable	Type of Injection	$\bar{x}$	SD	Me	Q_1_–Q_3_	W	*p*
Age (years)	-	37.8	7.5	38.0	33.0–43.0	0.976	0.534
Height (m)	-	1.66	0.06	1.65	1.63–1.70	0.970	0.331
Body mass (kg)	-	67.4	6.8	68.0	63.0–71.0	0.973	0.435
BMI	-	24.5	2.1	25.0	23.5–25.9	0.881	0.001
Scar length (cm)	PRP/placebo	6.7	0.3	6.5	6.5–7.0	0.732	<0.001
Density of injected agent (mL/cm)	PRP/placebo	0.20	0.03	0.21	0.19–0.22	0.949	0.063
Weight of resection (g)	PRP	670.2	247.9	675.0	498.0–810.0	0.969	0.322
	placebo	661.5	255.0	632.0	457.0–855.0	0.963	0.195
R after 3 m	PRP	104	8	102	99–110	0.987	0.914
	placebo	105	10	106	96–112	0.965	0.238
G after 3 m	PRP	72	9	71	67–75	0.975	0.483
	placebo	73	10	74	65–80	0.973	0.430
B after 3 m	PRP	78	9.61	78	72–84	0.978	0.597
	placebo	78	12	79	73–85	0.951	0.076
L after 3 m	PRP	30.52	3.51	30.29	28.71–32.04	0.983	0.803
	placebo	30.68	4.25	30.92	28.19–33.56	0.970	0.335
a after 3 m	PRP	20.48	2.49	20.49	18.72–22.04	0.964	0.221
	placebo	20.74	3.02	20.40	18.43–22.76	0.984	0.819
b after 3 m	PRP	4.21	2.20	3.90	2.72–5.94	0.967	0.279
	placebo	4.92	2.76	4.08	3.08–6.58	0.949	0.062
MI after 3 m	PRP	39.16	3.64	38.98	36.72–41.09	0.986	0.888
	placebo	38.79	4.20	38.92	35.73–42,42	0.970	0.353
EI after 3 m	PRP	16.18	2.79	16.09	14.80–17.24	0.929	0.014
	placebo	16.37	3.15	15.17	14.44–19.10	0.941	0.035
R after 6 m	PRP	115	16	116	106–123	0.960	0.160
	placebo	116	15	115	106–122	0.940	0.031
G after 6 m	PRP	82	17	80	71–91	0.963	0.206
	placebo	84	17	82	73–95	0.944	0.043
B after 6 m	PRP	89	16	87	79–97	0.977	0.559
	placebo	91	17	87	79–99	0.948	0.058
L after 6 m	PRP	34.99	6.36	34.75	30.58–38.55	0.971	0.359
	placebo	35.49	6.43	34.67	30.72–39.39	0.942	0.038
a after 6 m	PRP	20.72	3.52	21.06	17.57–22.93	0.971	0.357
	placebo	19.91	3.26	19.61	17.69–22.16	0.985	0.852
b after 6 m	PRP	4.89	2.95	4.06	2.21–6.97	0.940	0.032
	placebo	5.36	3.43	5.05	2.89–7.99	0.958	0.133
MI after 6 m	PRP	34.45	5.12	34.43	31.86–37.67	0.976	0.538
	placebo	34.57	5.21	35.21	32.01–38.26	0.962	0.178
EI after 6 m	PRP	15.22	3.95	14.94	12.31–17.69	0.961	0.175
	placebo	14.53	3.48	14.66	11.76–16.59	0.984	0.814
Scar assessment	PRP	4.0	0.7	4.0	4.0–4.5	0.844	<0.001
	placebo	4.0	0.7	4.0	4.0–4.5	0.886	0.001

x—mean, SD—standard deviation, Me—median, Q_1_–Q_3_—quartile range, W—Shapiro–Wilk statistics, *p*—probability for the Shapiro–Wilk test.

**Table 2 jcm-13-00691-t002:** Comparison of the average scores of scars 6 months after surgery for wounds injected with PRP and placebo.

Score	PRP (*n* = 41)	Placebo (*n* = 41)	Z/*p*
2	1	1	−0.01/ 0.996
3	6	6	
3.5	1	3	
4	22	18	
4.5	2	4	
5	9	9	

Mann–Whitney test, *n*—number of scars.

**Table 3 jcm-13-00691-t003:** Comparison of spectrophotometric parameters of scars between wounds injected with PRP and placebo and 3 and 6 months after surgery and for both variables.

Dependent Variables	Type of Injection (PRP vs. Placebo)	Time of Examination (3 m vs. 6 m)	Type of Injection × Time of Examination
R (B-C)	F = 0.32; *p* = 0.571	F = 44.16; *p* < 0.001	F = 0.01; *p* = 0.909
G (B-C)	F = 0.36; *p* = 0.553	F = 40.14; *p* < 0,001	F = 0.20; *p* = 0.656
B	F = 0.06; *p* = 0.808	F = 46.80; *p* < 0.001	F = 0.08; *p* = 0.777
L (B-C)	F = 0.07; *p* = 0.790	F = 49.18; *p* < 0.001	F = 0.14; *p* = 0.714
a	F = 0.22; *p* = 0.638	F = 0.69; *p* = 0.409	F = 2.20; *p* = 0.142
b (B-C)	F = 0.78; *p* = 0.381	F = 1.25; *p* = 0.266	F = 0.32; *p* = 0.572
MI	F = 0.02; *p* = 0.877	F = 53.33; *p* < 0.001	F = 0.16; *p* = 0.687
EI (B-C)	F = 0.18; *p* = 0.674	F = 13.65; *p* = 0.004	F = 1.03; *p* = 0.314

F—Fisher statistics, *p*—*p*-value, (B-C)—variables after Box–Cox transformation.

**Table 4 jcm-13-00691-t004:** Correlation between the scar’s score after 6 months (dependent variable) and different variables (age, BMI, weight of resected tissues, vertical scar length, the density of PRP/placebo) and spectrophotometric variables of the scar measured 3 months after the procedure in interaction with the type of injection (PRP/placebo) (GLM).

Variables	F	*p*
PRP/Placebo	0.56	0.458
Age	0.20	0.660
PRP/Placebo × Age	0.58	0.449
PRP/Placebo	2.03	0.158
BMI	2.27	0.136
PRP/Placebo × BMI	2.04	0.157
PRP/Placebo	0.84	0.362
Vertical scar length	8.81	0.004
PRP/Placebo × vertical scar length	0.84	0.362
PRP/Placebo	1.32	0.253
Density of injected PRP/Placebo	3.12	0.081
PRP/Placebo × Density of injected PRP/Placebo	1.35	0.249
PRP/Placebo	0.50	0.482
Weight of resected tissues	7.54	0.008
PRP/Placebo × Weight of resected tissue	0.60	0.442
PRP/Placebo	0.10	0.753
R after 3 m	4.67	0.034
PRP/Placebo × R after 3 m	0.09	0.765
PRP/Placebo	0.71	0.402
G after 3 m	4.32	0.041
PRP/Placebo × G after 3 m	0.69	0.407
PRP/Placebo	0.07	0.786
B after 3 m	3.12	0.081
PRP/Placebo × B after 3 m	0.07	0.785
PRP/Placebo	0.50	0.481
L after 3 m	4.05	0.048
PRP/Placebo × L after 3 m	0.50	0.481
PRP/Placebo	1.70	0.196
a after 3 m	0.44	0.510
PRP/Placebo × a after 3 m	1.72	0.194
PRP/Placebo	0.00	0.968
b after 3 m	5.81	0.018
PRP/Placebo × b po 3 m	0.05	0.816
PRP/Placebo	0.63	0.431
MI after 3 m	3.06	0.084
PRP/Placebo × MI after 3 m	0.65	0.424
PRP/Placebo	2.28	0.135
EI after 3 m	1.46	0.230
PRP/Placebo × EI after 3 m	2.33	0.131

**Table 5 jcm-13-00691-t005:** Multivariable analysis to evaluate the influence of independent variables on assessment of scars after 6 months (dependent variable).

Independent Variable	R^2^ = 0.2848, Corrected R^2^ = 0.1088; F (16.65) = 1.62 *p* < 0.089; Estimation Error: 0.67			
	b	SEb	t	*p*
PRP/Placebo	0.0436	0.1096	0.40	0.692
Age	−0.0094	0.1185	−0.08	0.937
Weight of resected tissues	0.0143	0.1756	0.08	0.935
Scar length	−0.1860	0.1757	−1.06	0.294
Density of injected PRP/Placebo	0.1717	0.1236	1.39	0.17
Body height	4.9669	1.8749	2.65	0.01
Body weight	−6.9170	2.6440	−2.62	0.011
BMI	5.9491	2.2878	2.60	0.012
R after 3 m (B-C)	0.1838	0.8551	0.21	0.83
G after 3 m (B-C)	0.5648	0.8150	0.69	0.491
B after 3 m	0.0133	0.4978	0.03	0.979
L after 3 m (B-C)	0.0200	0.5463	0.04	0.971
a after 3 m	−0.4391	0.6729	−0.65	0.516
b after 3 m (B-C)	0.2065	0.2559	0.81	0.423
MI after 3 m	0.2858	0.3263	0.88	0.384
EI after 3 m (B-C)	0.6677	0.7220	0.92	0.358

b—standardized regression coefficient, SEb—standard error for b, t—t test value, *p*—*p*-value.

## Data Availability

Data are available on request from the corresponding author.

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
