# Peer review of "Do Intraoperative Platelet-Rich Plasma Injections Influence the Final Appearance of Vertical Scars after Breast Reduction? Spectrophotometric Analysis"

_jcm, 2024, doi:10.3390/jcm13030691_

Round 1

Reviewer 1 Report

Comments and Suggestions for Authors

The study is well conducted and designed.

I suggest authors to mention in M&M if they had delayed healing at the T- inverted, since it is the most common complication described in the Wise pattern technique, and how did they manage it. They declared that complicated cases  where excluded from the study: what do you mean with complicated (please make a list of complication and also how often they occurred in your study population).

Author Response

Dear Editor and Reviewers,

Thank you for your interest in our manuscript entitled " Are intraoperative platelet-rich plasma injection and spectrophotometric parameters predictors of vertical scar’s final appearance in breast reduction?”. We would like to thank Reviewers for their valuable comments and reviews, which helped to improve the manuscript. In this revision we addressed all your comments (in the text changes are marked in red).

Sincerely,

The Authors

Reviewers’ comments:

Reviewer 1

The study is well conducted and designed.

Thank you!

I suggest authors to mention in M&M if they had delayed healing at the T- inverted, since it is the most common complication described in the Wise pattern technique, and how did they manage it.

Done. It was explained that “minor” problems in T-junction was not considered as complication, as it is rather more common than more uncommon. We presented our way to manage this. To note, this “problematic area” did not affect the results /not visible on photographs due to natural ptosis/ and spectrophotometric exam was done in the middle of the vertical scar. But, as you pointed on this problem, it would be worth and clinically very beneficial to focus further study on this small but troublesome area!

Inc.: “Delayed wound healing in T-junction without wound dehiscence over 5 mm, was not considered as complication due to its often appearance. In such cases Iodine dressing (Inadine) was administered for 7 days (with the recommendation of daily change) and then Triple Antibiotic Ointment was recommended for final epithelialization of the wound. In all cases on follow up visit after 2 weeks of such treatment T-junction wounds were closed and received routine recommendations.”

They declared that complicated cases where excluded from the study: what do you mean with complicated (please make a list of complication and also how often they occurred in your study population).

As suggested, we listed excluded cases due to complications: 3 - wound dehiscence (1 – in the middle of vertical scar and 2 – in both T-junctions, over 5 mm) and 2 – nipple areola complex necrosis (left sided, including about 1/3 of the upper pole of areola, treated with debridement and Iodine dressing (Inadine)).

Reviewer 2 Report

Comments and Suggestions for Authors

This study is quite intriguing; however, the title may be somewhat misleading, as it primarily focuses on assessing the effectiveness of Platelet-Rich Plasma (PRP) in the context of wound healing and scar appearance in breast reduction surgery, utilizing a comparative approach. Spectrophotometric parameters were employed to evaluate the scars.

It's noteworthy that PRP was administered only once intraoperatively, and as such, additional injections were not administered. This information should be explicitly outlined in the methods and discussion sections, given that PRP typically exhibits greater efficacy with repeated treatments.

Are there any additional data available regarding patients' comorbidities? Has the extent of resection (weight of resection) been considered in relation to the outcomes of the scars? Furthermore, have any postoperative complications impacted both the scars and spectrophotometric parameters?

I recommend conducting a multivariable analysis to control for confounding factors that may influence the primary outcomes. This will enhance the robustness of the study's findings.

Author Response

Dear Editor and Reviewers,

Thank you for your interest in our manuscript entitled " Are intraoperative platelet-rich plasma injection and spectrophotometric parameters predictors of vertical scar’s final appearance in breast reduction?”. We would like to thank Reviewers for their valuable comments and reviews, which helped to improve the manuscript. In this revision we addressed all your comments (in the text changes are marked in red).

Sincerely,

The Authors

Reviewers’ comments:

Reviewer 2

This study is quite intriguing; however, the title may be somewhat misleading, as it primarily focuses on assessing the effectiveness of Platelet-Rich Plasma (PRP) in the context of wound healing and scar appearance in breast reduction surgery, utilizing a comparative approach. Spectrophotometric parameters were employed to evaluate the scars.

Yes, there were two objectives of the study: 1 - to examine the effect of intraoperative injection of plate-let-rich plasma (PRP) on the final appearance of vertical scars after breast reduction and 2- verify spectrophotometric parameters as eventual predictors of scar’s aesthetic assessment. So, we used spectrophotometric parameters to look for “objective” differences between sides (apart from “subjective” scores) and as correlates of scars appearance. Because we identified early /after 3 months/ spectrophotometric parameter as a predictor of scar’s final assessment, we included this aspect in the title. But to make the title clearer (hopefully) it was changed to:

“Does intraoperative platelet-rich plasma injection influence vertical scar’s final appearance in breast reduction? Spectrophotometric analysis.”.

It's noteworthy that PRP was administered only once intraoperatively, and as such, additional injections were not administered. This information should be explicitly outlined in the methods and discussion sections, given that PRP typically exhibits greater efficacy with repeated treatments.

As requested, this was highlighted in the M&M section and discussed further in discussion. 

Inc. :” Additionally, the protocol of the study included only single, intraoperative injection of PRP while it would be worth to extend the protocol for multiple injections.”

Are there any additional data available regarding patients' comorbidities? Has the extent of resection (weight of resection) been considered in relation to the outcomes of the scars?

Yes, we included the missing information regarding comorbidities (hypertension in 3 patients). Also, weight of resection was reported /pls see table 1/and included in the analysis:

“Using the general linear model (GLM), it was checked whether the assessment of scars depends on the age and BMI of the patients, the extent of the procedure (the weight of the removed gland (= the weight of resection), the length of the scar), and the spectrophotometric parameters of the scar measured 3 and 6 months after the procedure” (pls see table 4)

Furthermore, have any postoperative complications impacted both the scars and spectrophotometric parameters?

Postoperative complication was an exclusion criterion, we listed excluded cases due to complications: 3 - wound dehiscence (1 – in the middle of vertical scar and 2 – in both T-junctions, over 5 mm) and 2 – nipple areola complex necrosis (left sided, including about 1/3 of the upper pole of areola, treated with debridement and Iodine dressing (Inadine)).

I recommend conducting a multivariable analysis to control for confounding factors that may influence the primary outcomes. This will enhance the robustness of the study's findings.

GLM analysis /with two-factor ANOVA/ and regression analysis to determine predictors of good and very good scar assessment were better explained. But also, according to your suggestion we performed a multivariable analysis to control for possible confounding factors. A table and a section were added:

3.3 Multivariable analysis

In the next step we performed a multivariable analysis to explain variability of scars’ assessment (after 6 months). The model including all variables appeared not to be significant ( p=0.089), although three variables (body weight, height and BMI) were significantly associated with scar’s assessment. Since many independent variables in the model were redundant (be-cause of strong inter-relation), in the next stage we built a stepwise model (limited to significant not inter-related variables). It included only one variable (R2= 0.1005, corrected R2= 0.0893; F(1.80)=8.94 p<0.0037; error: 0.67) that correlated with final scar’s assessment  - vertical scar’s length (b=-0.32, error for b = 0.106, t=-2.99, p=0.004). This variable explained about 10% of the assessment variability (Table 5).

Reviewer 3 Report

Comments and Suggestions for Authors

Congratulations to the authors for their work. Though the patient lot is not extensive, the study's design is sound, and its conclusions are interesting. The authors could expand their research in the future by applying platelet- rich plasma on mature scars.

Author Response

Dear Editor and Reviewers,

Thank you for your interest in our manuscript entitled " Are intraoperative platelet-rich plasma injection and spectrophotometric parameters predictors of vertical scar’s final appearance in breast reduction?”. We would like to thank Reviewers for their valuable comments and reviews, which helped to improve the manuscript. In this revision we addressed all your comments (in the text changes are marked in red).

Sincerely,

The Authors

Reviewers’ comments:

 Reviewer 3

Congratulations to the authors for their work. Though the patient lot is not extensive, the study's design is sound, and its conclusions are interesting. The authors could expand their research in the future by applying platelet- rich plasma on mature scars.

Thank you for your approval and the idea for future study!
